# AN INVESTIGATION OF DOMAIN GENERALIZATION WITH RADEMACHER COMPLEXITY

## ABSTRACT

The domain generalization (DG) setting challenges a model trained on multiple known data distributions to generalise well on unseen data distributions. Due to its practical importance, many methods have been proposed to address this challenge. However much work in general purpose DG is heuristically motivated, as the DG problem is hard to model formally; and recent evaluations have cast doubt on existing methods' practical efficacy – in particular compared to a well tuned empirical risk minimisation baseline. We present a novel learning-theoretic generalisation bound for DG that bounds unseen domain generalisation performance in terms of the model's empirical risk and Rademacher complexity – providing a sufficient condition for DG. Based on this insight, we empirically analyze the performance of several methods and show that their performance is indeed influenced by model complexity in practice. Algorithmically, our analysis suggests that tuning for domain generalisation should be achieved by simply performing regularised ERM with a leave-one-domain-out cross-validation objective. Empirical results on the DomainBed benchmark corroborate this.

## 1 INTRODUCTION

Machine learning systems have shown exceptional performance on numerous tasks in computer vision and beyond. However performance drops rapidly when the standard assumption of i.i.d. training and testing data is violated. This domain-shift phenomenon occurs widely in many applications of machine learning (14; 37; 25), and often leads to disappointing results in practical machine learning deployments, since data 'in the wild' is almost inevitably different from training sets.

Given the practical significance of this issue, numerous methods have been proposed that aim to improve models' robustness to deployment under train-test domain shift (37), a problem setting known as domain generalisation (DG). These span diverse approaches including specialised neural architectures, data augmentation strategies, and regularisers. Nevertheless, the DG problem setting is difficult to model formally for principled derivation and theoretical analysis of algorithms; since the target domain(s) of interest cannot be observed during training, and cannot be directly approximated by the training domains due to unknown distribution shift. Therefore the many popular approaches (37) are based on poorly understood empirical heuristics—a problem highlighted by (20), who found that no DG methods reliably outperform a well-tune empirical risk minimisation (ERM) baseline.

Our first contribution is to present an intuitive learning-theoretic bound for DG performance. Intuitively, while the held-out domain of interest is unobservable during training, we can bound its performance using learning theoretic tools similar to the standard ones used to bound the performance on (unobserved) testing data given (observed) training data. In particular we show that the performance on a held out target domain is bounded by the performance on known source domains, plus two additional model complexity terms, that describe how much a model can possibly have overfitted to the training domains. This provides a *sufficient* condition for DG and leads to several insights.

Firstly, our theory suggests that DG performance is influenced by a trade-off between empirical risk and model complexity that is analogous to the corresponding and widely understood trade-off that explains generalisation in standard i.i.d. learning as an overfitting-underfitting trade-off (17). Based on this, we conjecture that the efficacy of the plethora of available strategies (37) – from data-augmentation to specialised optimisers – is largely influenced by explicitly or implicitly choosing

different fit-complexity trade-offs. And further, that the importance of proper tuning as discussed by (20) may be mediated in part by the impact of hyperparameters on model complexity.

Analyzing these issues empirically is difficult, as model complexity is hard to carefully control in deep learning due to the large number of relevant factors (explicit regularisers, data augmentation, optimiser parameters, etc). (20) attempted to address this by random hyper-parameter search in the DomainBed benchmark, but are hampered by the computational infeasibility of accurate hyper-parameter search. In this paper, we use linear models, random forests, and shallow MLPs to demonstrate more clearly how cross-domain performance depends on model complexity.

Secondly, our theory further suggests that the model selection criterion (22) is an important factor in DG performance. In particular, regularisation should be stronger when optimizing for future DG performance than when optimizing for performance on seen domains, which we confirm empirically. Further, our theoretical and empirical results show that, contrary to the conclusion of (20), domain-wise cross-validation is a better objective to drive DG model selection than instance-wise.

In summary, the take-home messages of our analysis are: (i) When achievable, low empirical risk combined with low complexity provides a *sufficient condition* for DG. (ii) Model fit vs complexity trade-off is a key factor in practical DG performance. (iii) The complexity control strategy used to determine bias-variance trade-off is crucial, with peak DG performance achieved when optimizing model complexity based on domain-wise validation. (iv) The regularisation strength required for optimal DG is greater than for conventional optimization for within-domain performance.

## 2 RELATED WORK

**Theoretical Analysis of the DG Setting and Algorithms**   The DG problem setting was first analysed in (9). Since then there have been some attempts to analyse DG algorithms from a generalisation bound perspective (29; 8; 23; 2; 33). However these studies have theoretical results that are either restricted to specific model classes, such as kernel machines, or make strong assumptions about how the domains seen during training will resemble those seen at test time—e.g., that all domains are convex combinations of a finite pre-determined set of prototypical domains. In contrast, our Rademacher complexity approach can be applied to a broad range of model classes (including neural networks), and makes comparatively milder assumptions about the relationship between domains—i.e., they are i.i.d. samples from another arbitrary distribution over domains.

The majority of the existing work investigating the theoretical foundations of DG follow the initial formalisation of the domain generalisation problem in (9), where the goal is to minimise the expected error over unseen domains. However, several recent works have also explored the idea of bounding the error on a single unseen domain with the most pathological distribution shift (3; 24). This type of analysis is typically rooted in methods from causal inference, rather than statistical learning theory. As a consequence, they are able to make stronger claims for the problems they address, but the scope of their analysis is necessarily limited to the scenarios where their assumptions about the underlying causal structures are valid. For example, (24) provides bounds that assume problems conform to a specific class of structural equation models, and the analysis assumes that infinite training data is available within each observed training domain. We primarily address the standard DG formalisation of (9), which is concerned with the expected performance of a model on new domains sampled from a distribution over domains. However, we also provide a means to transform any bound on the expected (or "average-case") risk to a high-confidence bound on the worst-case risk. Possibly the most similar work to our theoretical contributions is (1), that also provides learning-theoretic generalisation bounds for DG. However, their analysis only applies to finite hypothesis classes (which does not include, e.g., linear models or neural networks), whereas ours can be applied to any class amenable to analysis with Rademacher complexity.

**Empirical Analysis**   The main existing empirical analysis on DG is (20), who compared several state of the art methods using DomainBed, a common benchmark and hyper-parameter tuning protocol. They ultimately defend Empirical Risk Minimization (ERM) over more sophisticated alternatives on the grounds that no competitor consistently beats it. We also broadly defend ERM, and build on the same benchmark, but differently we provide a deeper analysis into when and why ERM works. More specifically: (i) We provide a theoretical analysis of ERM's generalisation quality unlike the prior purely empirical evaluation, (ii) We re-use the DomainBed benchmark to directly

corroborate this theory under controlled conditions using linear models and random forests, where model complexity can be more feasibly tuned. (iii) Our complexity analysis provides a possible explanation of the hyperparameter sensitivity observed by (20), and possible explanation for why ERM is hard to beat (if methods primarily influence complexity). (iv) We identify, and empirically validate, the preferred model selection criterion for DG, a point which was inconclusive in (20).

## 3 BOUNDING RISK FOR DOMAIN GENERALIZATION

**I.i.d. learning** is concerned with learning a mapping from some input space $\mathcal{X}$, to a label space $\mathcal{Y}$, given data drawn from a distribution on $\mathcal{X} \times \mathcal{Y}$. One aims to find a model, $f^* \in \mathcal{F}$, that minimises the expected loss (also called risk) on unseen data,

$$f^* = \arg\min_{f \in \mathcal{F}} L_p(f), \qquad L_p(f) = \mathbb{E}_{(\vec{x},y) \sim p}[\ell(f(\vec{x}), y)], \tag{1}$$

where $p$ is the data distribution and $\ell(\cdot, \cdot)$ is the loss function. In practice, we only have access to a finite set of data, $S = \{(\vec{x}_i, y_i)\}_{i=1}^m$, sampled i.i.d. from this distribution, so must minimise an empirical risk estimate,

$$\hat{f} = \arg\min_{f \in \mathcal{F}} \hat{L}_p(f), \qquad \hat{L}_p(f) = \frac{1}{m} \sum_{i=1}^m \ell(f(\vec{x}_i), y_i). \tag{2}$$

where $m$ is the number of training examples. A central focuses of statistical learning theory is to bound the difference between these two types of risk. For example, in the standard single domain setting with a loss function bounded between zero and one (e.g., the misclassification rate) this can be done via

$$L_p(f) \leq \hat{L}_p(f) + 2\hat{\mathcal{R}}_m(\mathcal{F}) + 3\sqrt{\frac{\ln(2/\delta)}{2m}}, \tag{3}$$

which holds with probability at least $1 - \delta$, and $\hat{\mathcal{R}}(\mathcal{F})$ is known as the empirical Rademacher complexity of the hypothesis class, $\mathcal{F}$. This complexity term is defined as

$$\hat{\mathcal{R}}_m(\mathcal{F}) = \mathbb{E}_{\vec{\sigma}}\left[\sup_{f \in \mathcal{F}} \frac{1}{m} \sum_{i=1}^m \sigma_i \ell(f(\vec{x}_i), y_i)\right], \tag{4}$$

where $\sigma_i$ are Rademacher random variables, distributed such that $P(\sigma_i = -1) = P(\sigma_i = 1) = 0.5$. For a hypothesis class consisting of norm-constrained linear classifiers or depth-restricted decision trees (or ensembles of decision trees), one can achieve a bound on Rademacher complexity that scales as $\mathcal{O}(m^{\frac{-1}{2}})$[1]. There are a variety of ways to define hypothesis classes for neural networks, but most recent approaches fix an architecture and specify constraints on the norms of weights or distances they can move from their initialisations (4; 30; 19).

**Domain generalisation.** While standard i.i.d. learning assumes all data come from the same distribution, the DG problem setting assumes the existence of an *environment* $\mathcal{E}$, of distributions $p$ (9). Throughout the remainder we assume all probability density functions in supp($\mathcal{E}$) are measurable according to some common base measure. Note that we do not restrict what types of differences one could see between different domains sampled from $\mathcal{E}$: for any two distributions in $p, q \in \text{supp}(\mathcal{E})$ it could be the case that either $p(\vec{x}) \neq q(\vec{x})$, or $p(y|\vec{x}) \neq q(y|\vec{x})$, or even that both types of distribution shift have occurred. The conceptually simplest—and often implicitly assumed—goal of DG methods is to minimise the expected risk across different distributions that could be sampled from the environment,

$$L^{\mathcal{E}}(f) = \mathbb{E}_{p \sim \mathcal{E}}[L_p(f)]. \tag{5}$$

This object is also the most commonly analysed idealised objective in the learning theory literature (9; 29; 8; 33), but other formulations also exist (3). As with the single domain learning problem, we only have access to an empirical estimate of the risk,

$$\hat{L}^{\mathcal{E}}(f) = \frac{1}{n} \sum_{j=1}^n \hat{L}_{p_j}(f), \tag{6}$$

where we assume for ease of exposition that all $n$ domains have the same number of examples.

---

[1]See Section 26.2 of (34) for a textbook example of bounding the Rademacher complexity of linear models with bounded weight norms.

## 3.1 Target Domain-Dependent Generalisation Bound

One can construct a bound on the error of a model operating on a novel domain, $q$, by measuring the mean total variation distance between $q$ and each source domain $p_i$, as follows.

**Theorem 1.** *For a 1-Lipschitz loss, $\ell(\cdot, \cdot)$, taking values in $[0, 1]$, with confidence at least $1 - \delta$ for all $f \in \mathcal{F}$ we have that*

$$L_q(f) \leq \hat{L}^{\mathcal{E}}(f) + 2\hat{\mathcal{R}}_{mn}(\mathcal{F}) + \frac{2}{n}\sum_{i=1}^{n} \|q - p_i\|_{TV} + 3\sqrt{\frac{ln(2/\delta)}{2mn}}, \qquad (7)$$

*where $n$ is the number of training domains and $m$ is the number of training examples in each domain.*

The proof can be found in the appendix. While such bounds at first glance seem intuitive and appealing, they are of limited practical use in the DG setting, because it is *not possible to either measure or change the TV distance between the source and target distributions* in the DG setting.

## 3.2 Target Domain-Independent Generalisation Bound

We next bound the generalisation gap between the observed empirical risk, $\hat{L}^{\mathcal{E}}$, on the source domains and the expected risk, $L^{\mathcal{E}}$, on unseen domains that holds uniformly for all hypotheses in $\mathcal{F}$.

**Theorem 2.** *For a 1-Lipschitz loss, $\ell(\cdot, \cdot)$, taking values in $[0, 1]$, with confidence at least $1 - \delta$ for all $f \in \mathcal{F}$ we have that*

$$L^{\mathcal{E}}(f) \leq \hat{L}^{\mathcal{E}}(f) + 2\hat{\mathcal{R}}_{mn}(\mathcal{F}) + 2\hat{\mathcal{R}}_n(\mathcal{F}) + 3\sqrt{\frac{ln(4/\delta)}{2mn}} + 3\sqrt{\frac{ln(4/\delta)}{2n}}, \qquad (8)$$

*where $n$ is the number of training domains and $m$ is the number of training examples in each domain.*

The proof can be found in Appendix A. We remark that this theorem makes no assumption on the distribution over domains $\mathcal{E}$, model invariances, or similarity between source and target domain. The key assumption—shared with (9)—is that source and target are independent samples from $\mathcal{E}$.

**Discussion** Theorem 2 tells us that expected risk on unseen domains is bounded by the empirical risk (training loss) on seen domains, plus Rademacher complexity terms and sampling error terms that decay with the number of domains and training instances per domain. As with typical single-domain bounds, the latter sampling error terms are not under control of the model designer. The former Rademacher terms describe the complexity of the chosen model class and govern how much it could possibly overfit to the seen domains while minimising empirical risk. In the case of linear models, these terms depend on weight norms, while in the case of deep models they further depend on properties of the chosen network architecture. Mirroring conventional generalisation in standard i.i.d. learning, a very simple model may minimise the Rademacher terms, $\hat{\mathcal{R}}$, while producing high empirical risk, $\hat{L}^{\mathcal{E}}$, and vice-versa. Thus good generalisation critically depends on a carefully chosen empirical risk vs model complexity trade-off. The difference between Theorem 2 and single domain bounds (e.g., Equation 3) is the additional dependence on the number of domains $n$ and additional Rademacher complexity term. This highlights an important insight: When the goal is to generalise to new domains, the risk of overfitting is higher. *Therefore a lower complexity model is optimal for held out domain performance compared for seen domain performance in standard i.i.d. learning.*

## 3.3 Bounding the Excess Risk of ERM

As a second theoretical contribution, we next bound the excess expected risk between the ERM solution $\hat{f}$ and the *best possible model $f^*$* within the function class $\mathcal{F}$. Note that bounding excess risk, as opposed to the generalisation gap, overcomes some of the known issues with theoretically analysing the generalisation properties of deep networks (7).

**Corollary 1.** *With probability at least $1 - \delta$, the excess risk of the empirical risk minimiser in $\mathcal{F}$ is bounded as*

$$\mathbb{E}[L^{\mathcal{E}}(\hat{f}) - L^{\mathcal{E}}(f^*)] \leq 2\hat{\mathcal{R}}_{mn}(\mathcal{F}) + 2\hat{\mathcal{R}}_n(\mathcal{F}) + 2\sqrt{\frac{ln(4/\delta)}{2mn}} + 2\sqrt{\frac{ln(4/\delta)}{2n}}.$$

The proof (in Appendix B) uses the same technique as the usual proof for excess risk (see, e.g., (31)), and some of the intermediate results required for Theorem 2.

**Discussion**  Corollary 1 tells us that the gap between ERM and the *best possible* predictor in the function class depends on the same complexity terms observed in Theorem 2. In particular, for any typical hypothesis class, ERM converges to the optimal classifier at a rate of $\mathcal{O}(1/\sqrt{mn}) + \mathcal{O}(1/\sqrt{n})$. To justify itself theoretically, any future DG method that claims to be better than ERM should either: (1) demonstrate a faster convergence rate than this—at least by an improved constant factor; or (2) formally show that the chosen hypothesis class is composed of models that can extrapolate to new domains without additional data. The latter would likely involve making specific assumptions about the underlying data generation process, coupled with analysis of a specific hypothesis class using methods from causal inference. An example of such analysis is given by (24). Methods based on causal inference have the potential to give bounds with much better convergence rates than the rate given above. However, because one must make assumptions about the underlying family of structural equation models, the applicability of such bounds is much more restricted than our Rademacher complexity technique, which does not require these assumptions.

### 3.4  FROM AVERAGE-CASE TO WORST-CASE

For some applications—such as safety critical systems that cannot be allowed to fail—one is interested in the bounding the error in the worst-case domain,

$$\tilde{L}^{\mathcal{E}}(f) = \sup_{p \in \text{supp}(\mathcal{E})} L_p(f). \tag{9}$$

Some papers accomplish this by making assumptions about the distribution $\mathcal{E}$. E.g., by explicitly constructing it as a convex combination of known domains, or showing that all models in $\mathcal{F}$ are "invariant" to all domains in the support (3; 24). We instead aim to construct a bound that will hold for most—but not necessarily all—domains that could be seen at test time. This allows us to instead only make a relatively mild assumption about bounded variance, as shown in the following result.

**Lemma 1.** *For $p \sim \mathcal{E}$, the following holds with confidence at least $1 - \kappa$,*

$$L_p(f) \le L^{\mathcal{E}}(f) + \sqrt{\frac{1-\kappa}{\kappa} Var_{q \sim \mathcal{E}}[L_q(f)]}.$$

*Proof.* The statement follows by aggregating performance on each domain into a scalar (from the definition of $L_q(\cdot)$), then computing the variance via the law of the unconscious statistician. Finally, an application of the confidence interval formulation of a one-sided variant of Chebyshev's inequality is used. This concentration inequality is also sometimes known as Cantelli's inequality. □

As $\kappa$ goes to zero, this Lemma allows us to bound the error for individual domains in increasingly larger subsets of the support of $\mathcal{E}$, thus achieving something practically similar to bounding Equation 9. Interestingly, those methods that attempt address worst-case domain generalisation by proving invariance of models to the underlying domain (i.e., those that show $L_p(f) = L_q(f)$, $p \neq q$) will have zero variance, by definition. Thus, the average-case bound in Theorem 2 also provides a bound on the worst-case generalisation error in the case of invariant hypothesis classes.

For those methods that do not rely on domain expertise to prove invariance, it should be noted that minimising the sample variance on the available training domains will not necessarily lead to minimising the population variance in Lemma 1. However, it is possible to construct an upper bound to the population variance, yielding the worst-case bound below.

**Theorem 3.** *For $p \sim \mathcal{E}$, the following holds with confidence at least $1 - (\delta + \kappa)$,*

$$L_p(f) \le A + \sqrt{\frac{1-\kappa}{\kappa} A},$$

*where $A$ is a bound on $L^{\mathcal{E}}(f)$ that holds with confidence $1 - \delta$ (e.g., from Theorem 2).*

The proof for this result can be found in Appendix C. This result suggests that one can apply a simple transform to the average-case bound provided by Theorem 2 in order to obtain a high-confidence upper bound on the worst-case error in unseen domains.

### 3.5 From Bounds to Algorithms

The analyses above suggest that to optimise for performance on held-out domains, one should simultaneously minimise empirical risk and model complexity. In practice the simplest way to tune complexity is via regularisation. To find the regularisation hyperparameters that work best for the average-case performance, one should measure *held out domain* performance via cross-validation (if there are few domains available), or with a single validation set of held-out domains (if many domains are available). This provides an unbiased estimate of the expected risk, $L^{\mathcal{E}}$, (denoted as **domain-wise** CV criterion). In contrast, optimising for performance on held out instances of known domains as recommended by (20) (denoted **instance-wise**) is an optimistically biased estimator.

If worst-case performance is of interest, one typically cannot obtain an unbiased estimate of $\tilde{L}^{\mathcal{E}}$. However, it is interesting to note that the link between Theorems 2 and 3 somewhat surprisingly suggests that any strategy that works well for the average-case objective also work for the worst-case objective. This is because, although the worst case bound is more pessimistic, both bounds will be minimised for the same regulariser strength (Rademacher complexity); the transform from average-case to worst-case bound is a monotonically increasing function, so the location of extreme points (i.e., optimal fit-complexity trade-off) is preserved. Thus a novel insight is that we can use the single domain-wise validation strategy above to optimise for both average and worst-case DG performance.

## 4 Experiments

Based on our previous theoretical analysis, we conduct experiments on DomainBed (20) and we study the following questions: (1) Theorem 2 shows that novel domain generalisation is governed by empirical-risk complexity trade-off. *Can we directly demonstrate the dependence of generalisation on complexity by controlling model complexity?* (2) *Could the hyperparameter sensitivity of DG methods observed by (20) be mediated by model complexity?* (3) Given that model complexity is a key determinant of generalisation, *what is the best objective for tuning regularisation strength?* (4) *Could our bounds ever generate non-trivial guarantees for cross-domain recognition performance?*

### 4.1 Domain Generalisation Performance Depends on Model Complexity

To directly investigate the impact of model complexity on domain generalisation, we use pre-computed deep features [2] and shallow classifiers linear SVM and Random Forest (RF). For linear models, tight bounds on model complexity are known and can be directly controlled by a scalar hyperparameter (5). The objective is convex so confounding factors in deep network training (optimiser choice, early stopping, etc) disappear, and training is fast enough that we can densely and exhaustively evaluate a wide range of complexities. Similarly for RF, complexity is controlled by the depth of the trees when the models are applied to binarised features (10). This is a direct consequence of the resulting hypothesis class being finite, and an application of Massart's lemma (5).

**Setup**  We use DINO (12) pre-trained DINO-ViTB8 as deep features, and then train classifiers using scikit-learn. We experiment on six different DG benchmarks, including RotatedMNIST (18), PACS (26), VLCS (16), OfficeHome (36), SVIRO (15) and Terra Incognita (6). For linear SVM, we use the objective $\frac{C}{n}\sum_i^n \ell(f_w(x_i), y_i) + \|w\|^2$, with parameter $C$ controlling the complexity of model $w$ through loss-regularisation trade-off, and search a range of $\log_2 C$ in $\{-10, \ldots, 10\}$. For RF we search a range of depths $D$ in $\{1, \ldots, 30\}$. We then conduct two experiments, holding training and test split size constant across both: (i) Training on the train splits of all domains, and testing on the test splits of all domains (i.e., standard i.i.d. learning). (ii) Training on the train splits of three domains, and testing on the test splits of a held out domain (i.e., DG performance).

**Results**  The results in Fig. 1 average over 5 random seeds for dataset splitting, and all choices of held-out target domain. From these we can see that: (i) Experiments exhibit the classic trade-off between fitting the data well and constraining hypothesis class complexity appropriately. For SVM we observe underfitting for high regularisation (small C), and overfitting at low regularisation (large C). RF is similar but with less severe overfitting. (ii) Comparing the within- and across-domain evaluation: across-domain leads to lower performance—which is to be expected, due to the distribution shift.

---

[2]Note that using a *fixed* feature extractor trained on independent data does not impact the model complexity or associated generalisation bound.

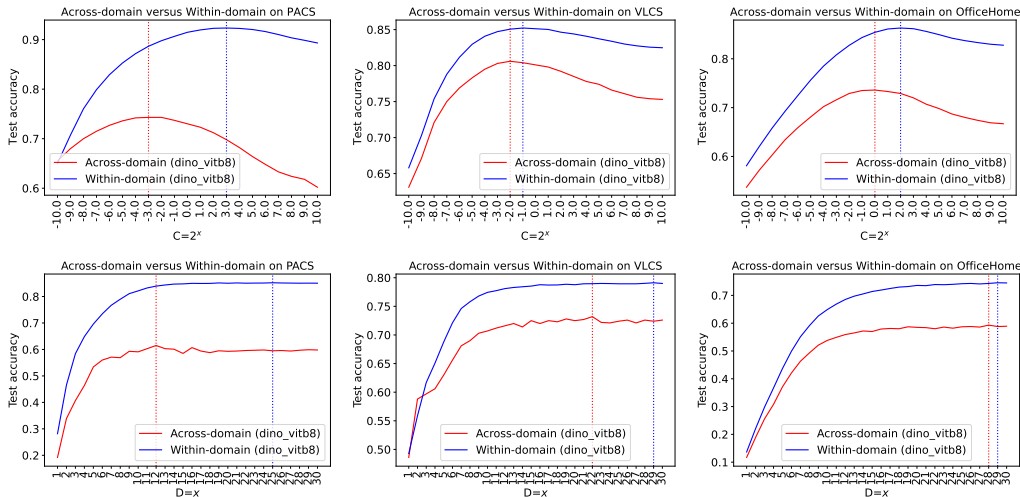

Figure 1: Linear SVM (Top) and Random Forest (Bottom) performance on DomainBed benchmark datasets. Optimal tuning for performance on novel target domains (red) always requires stronger regularisation (lower $C$ or $D$) than for performance on seen domains (blue).

A more noteworthy observation is that, the optimal regularisation for novel-domain performance is stronger than for seen-domain performance (red vertical lines left of blue). This corroborates our theoretical result that the ideal model complexity is lower for DG than for conventional i.i.d. learning. More examples are in Appendix Fig. 4. Note that these results are an exhaustive search of model complexity, and do not constitute a complete algorithm, which would need to chose a specific complexity. We evaluate this in Section 4.3.

## 4.2 Impact of Hyper-parameters on Complexity and DG Performance

Our experiment in Section 4.1 showed that linear models exhibit a very clean trade-off between complexity and generalization. However there are many subtle factors that influence effective model complexity in neural networks besides architecture: learning rate, learning rate schedule, weight decay, early stopping, etc. The prior evaluation (20) attempted to control these factors for state of the art neural DG methods by applying a common hyperparameter search strategy that aimed to find the best tuning (cf: Fig. 1) for each competitor. Under these conditions, they found that existing state of the art methods failed to outperform well tuned ERM. We observe that many of these factors controlled by DomainBed directly (e.g., weight decay) or implicitly (e.g., batch size) impact model complexity. To study such hyper-parameter tuning through the lens of complexity, we take one representative factor of stopping iteration, and measure its impact on model complexity – as a possible explanation of why it is important to control.

**Setup** To ensure that complexity can be measured well, we work with 2-layer MLPs. Specifically, we take fixed ImageNet pre-trained ResNet-18 features, and feed them to MLPs, which are then trained using the DomainBed framework. We then report the results of an ERM model, checkpointed at a range of training iterations.

**Measuring Model Complexity** Because we have limited our attention to 2-layer MLPs, we can take advantage of a model capacity metric that is specialised for this class of models. We retrospectively determine the complexity of a trained network using the measure proposed by (30), which bounds the Rademacher complexity of a 2-layer MLP hypothesis classes. More concretely, the expression used for computing complexity is $\|V\|_F(\|U - U^0\|_F + \|U^0\|_2)$, where $U$ is the weight matrix of the first layer, $U^0$ is its random initialisation, and $V$ is the weight matrix of the second layer. We use $\|\cdot\|_F$ to denote Frobenius norm, and $\|\cdot\|_2$ to indicate the spectral norm. Note that for simplicity we have omitted constant factors that depend only on the architecture and problem setting, and not the learned weights, as we use the same architecture for all methods we investigate.

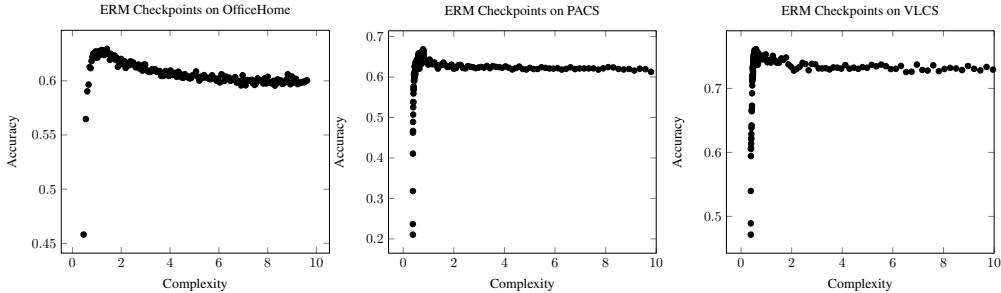

Figure 2: Leave-one-domain out cross-validation performance of shallow neural networks as a function of their measured model complexity while training on DomainBed (DB) when checkpointed every 300 iterations up to 50,000. The overall results are consistent with a standard bias-variance trade-off (cf. Fig 1 for linear models), with peak performance in the middle of the complexity range.

**Results** The results in Fig. 2 summarise the trade-off between measured model complexity (x-axis), and unseen domain test accuracy (y-axis), averaged over all held out domains for each dataset for an ERM model checkpointed every 300 iterations against the corresponding complexity. Neural models typically gain complexity with iterations (32; 21), and we see this in the plot. The overall result is one of overfitting-underfitting trade-off curves mirroring Fig. 1. This explains why checkpoint selection was found to be crucial in (20) – it directly affects complexity and hence generalisation via Thm 2. More generally, to the extent that all the other considered hyper-parameters influence complexity (we leave quantifying this to future work), it may explain why hyper-parameter selection more generally is crucial to control for DG (20). A limitation of this analysis is that we do not have tight bounds for deep neural networks, so can not repeat this same analysis for popular models such as ResNet.

## 4.3 EVALUATING LINEAR MODELS ON DOMAINBED

Having shown that model complexity is a major factor in DG performance, we next consider practical algorithms for complexity tuning. From Fig. 1, we see within-domain and cross-domain evaluation have different optimal tuning. When automating the search for a good regularisation parameter, these evaluation criteria correspond respectively to hyperparameter selection based on a validation set from the seen source domains (**instance-wise**), vs a validation set drawn from a held out source domain (**domain-wise**). As discussed in Section 3.5, the latter criterion corresponds to a unbiased estimator of expected cross-domain generalisation performance $L^{\mathcal{E}}(f)$, which our theorem bounds.

**Setup** We performed DG evaluation on DomainBed using ERM with linear models using DINO features as in Section 4.1 and pre-trained ResNet-50 for direct comparability to (20). For each held out domain, we performed hyperparameter tuning with either instance-wise cross-validation (where validation folds are drawn from the same source domains used for training) or domain-wise cross-validation (where validation folds are drawn from held-out domains not seen during training).

**Results** The results in Tab. 1 report the average accuracy across held out domains, and the average selected $\log C$ regularisation strength (See Appendices for the full breakdown of results for each held-out domain). The domain-wise cross-validation objective leads to similar or better accuracy, and similar or smaller C value selection (i.e., stronger regularisation). This outcome is as predicted by our theory, but different to the conclusion of (20). We attribute this difference to the difficulty of accurately tuning tuning stochastically trained neural networks as attempted in (20).

## 4.4 INSTANTIATING THE BOUND

A limitation of our theory, shared by most existing work on statistical learning theory, is that – while derived bounds give insights into the factors affecting generalization such as complexity – they are often not tight enough to provide useful performance guarantees for many problems and models of interest. As a counterpoint to this we illustrate one example where our bound provides non-trivial guarantees. While benchmarks within DomainBed have a relatively small number of domains, many problems in the broader AI community have a larger number of domains, making them more amenable to guarantees. For example, the LEAF (11) suite of benchmarks popular in the Federated

Table 1: Comparing model selection criteria using linear SVMs. Domain-wise validation usually outperforms instance-wise and selects stronger regularisation (lower $C$) as predicted by our theory.

| | DINO VIT-B8 | | | | ResNet-50 | | | |
| | Domain Wise | | Instance Wise | | Domain Wise | | Instance Wise | |
| | Acc | $\log C$ | Acc | $\log C$ | Acc | $\log C$ | Acc | $\log C$ |
| --- | --- | --- | --- | --- | --- | --- | --- | --- |
| PACS | 74.3 | -1.73 ($\pm$0.40) | 70.6 | 1.56 ($\pm$0.66) | 68.1 | 2.43 ($\pm$0.69) | 66.9 | 5.03 ($\pm$0.66) |
| VLCS | 80.4 | -1.04 ($\pm$0.69) | 80.5 | -0.87 ($\pm$0.35) | 76.7 | 1.73($\pm$1.20) | 77.6 | 2.25 ($\pm$0.35) |
| OfficeHome | 73.6 | -0.17 ($\pm$0.35) | 73.1 | 1.04 ($\pm$0.40) | 70.0 | 1.91 ($\pm$0.35) | 69.1 | 3.64 ($\pm$0.66) |
| SVIRO | 97.2 | 0.21 ($\pm$0.33) | 95.3 | 4.85 ($\pm$0.46) | 93.3 | 4.16($\pm$0.92) | 92.7 | 6.93 ($\pm$0.00) |
| Terra Inc. | 45.4 | -0.17 ($\pm$0.87) | 38.8 | 5.37 ($\pm$0.35) | 28.3 | 2.08 ($\pm$1.27) | 26.9 | 6.93 ($\pm$0.00) |

Learning community have a larger number of domains. We consider the multi-domain text analysis task defined in LEAF. The task is binary sentiment classification of tweets from different users, with each user defining a domain. We train a random forest on this task, and show the guaranteed DG classification accuracy along with the empirical DG classification accuracy in Figure 3.

## 5 CONCLUSION

From the results we can see that our Theorem 2 (together with the RF bound from (10)) provides a mostly non-trivial (better than random performance) guarantee for small depth RFs. We studied the performance of domain generalisation methods through the lens of model complexity (bias-variance trade-off) from both theoretical and empirical perspectives. Both perspectives show that complexity impacts cross-domain generalisation in a way that mirrors the bias-variance trade-off in conventional i.i.d. learning—but where stronger regularisation is required if optimising for cross-domain generalisation than if optimising for conventional within-domain generalisation. We clarified the preferred model selection criterion in each case.

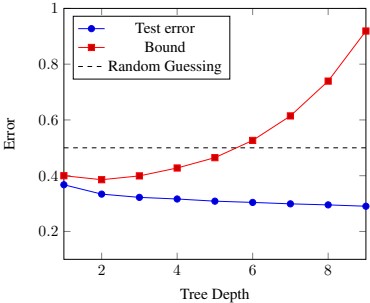

Figure 3: Empirical and guaranteed DG performance on LEAF Sentiment analysis using RF.

We set out to address four questions, which can be answered as: (1) Yes. The influence of cross-domain generalisation on complexity can be directly demonstrated using linear models and RFs. (2) Maybe. Hyperparameter choice in DomainBed influences neural network complexity, and this may in turn influence neural generalisation. (3) If the goal is to optimise for domain generalisation, then domain-wise validation is preferred to instance-wise validation for model-selection. This will lead to stronger regularisation for better out-of-domain generalisation (at a cost to in-domain generalisation). (4) Yes. While current complexity bounds may not be tight enough to provide useful guarantees for DomainBed and deep neural networks, many benchmarks and models popular in the broader AI community can already support non-trivial performance guarantees for cross domain generalisation.

Using general purpose pre-trained features (12) as we did here, and focusing on learning shallow models that can be *accurately* tuned for DG may be a promising alternative avenue for improving DG in practice compared to popular focus on sophisticated neural DG methods (37). Although achieving state of the art performance is not our focus, we note that our results in Tab. 1 are not far behind end-to-end trained state of the art (20), despite using fixed features and shallow models.

While we focused on ERM, we remark that our bound applies to any class of models for which Rademacher complexity can be computed—our analysis is rooted in uniform convergence, so it is agnostic to the training algorithm used to obtain that model. For example, the choice of auxiliary loss function, whether (9) or not (13) the training algorithm uses domain labels, or whether some form of adversarial training is employed (35) will not have any bearing on our bounds unless they also change the underlying hypothesis class in some way.

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

## A  PROOF OF MAIN RESULTS

We use a slightly modified version of the standard empirical Rademacher complexity bound on generalisation error, as stated by (28) but originally shown by (5), where the modification is to weaken the i.i.d. assumption to be only an independence assumption—i.e., allow instances to be drawn from different distributions. The proof is exactly the same (because McDiarmid's inequality requires only independence, and not identical distributions), but we re-state it here for completeness.

**Theorem 4.** *For $p_1, ..., p_n$ independent samples from $\mathcal{E}$, and 1-Lipschitz loss $\ell(\cdot, \cdot)$ taking values in $[0, 1]$, the following holds with confidence at least $1 - \delta$,*

$$\frac{1}{n}\sum_{j=1}^{n} L_{p_j}(f) \leq \frac{1}{n}\sum_{j=1}^{n} \hat{L}_{p_j}(f) + 2\hat{\mathcal{R}}_{mn}(\mathcal{F}) + 3\sqrt{\frac{ln(2/\delta)}{2mn}}, \tag{10}$$

*where $\hat{L}_{p_j}(f)$ is measured on $S_{p_j} = \{(\vec{x}_{ij}, y_{ij})\}_{i=1}^{m}$, a collection of $m$ i.i.d. samples from $p_j$.*

*Proof.* Let $\mathcal{S} = S_{p_1} \bigcup ... \bigcup S_{p_n}$ and define

$$\Phi(\mathcal{S}) = \sup_{f \in \mathcal{F}} \frac{1}{n}\sum_{j=1}^{n} (L_{p_j}(f) - \hat{L}_{p_j}(f)). \tag{11}$$

Note that $\Phi(\mathcal{S})$ satisfies the bounded differences property required by McDiarmid's inequality: i.e., if we construct $\mathcal{S}'$ by replacing any one of the $(x_{ij}, y_{ij})$ in $\mathcal{S}$ with another random variable also drawn from $p_j$, then $|\Phi(\mathcal{S}) - \Phi(\mathcal{S}')| \leq \frac{1}{mn}$. Therefore McDiarmid's inequality implies that with confidence at least $1 - \frac{\delta}{2}$

$$\Phi(\mathcal{S}) \leq \mathbb{E}_{S_{p_{1:n}} \sim p_{1:n}}[\Phi(\mathcal{S})] + \sqrt{\frac{\ln(2/\delta)}{2mn}}. \tag{12}$$

We continue by bounding the expected value of $\Phi(\mathcal{S})$,

$$\mathbb{E}_{S_{p_{1:n}} \sim p_{1:n}}[\Phi(\mathcal{S})] \tag{13}$$

$$= \mathbb{E}_{S_{p_{1:n}} \sim p_{1:n}}\left[\sup_{f \in \mathcal{F}} \frac{1}{n}\sum_{j=1}^{n}(L_{p_j}(f) - \hat{L}_{p_j}(f))\right] \tag{14}$$

$$= \mathbb{E}_{S_{p_{1:n}} \sim p_{1:n}}\left[\sup_{f \in \mathcal{F}} \frac{1}{n}\sum_{j=1}^{n}\left(\mathbb{E}_{S'_{p_j} \sim p_j}\left[\frac{1}{m}\sum_{i=1}^{m}\ell(f(\vec{x}'_{ij}, y'_{ij})\right] - \frac{1}{m}\sum_{i=1}^{m}\ell(f(\vec{x}_{ij}, y_{ij})\right)\right] \tag{15}$$

$$\leq \mathbb{E}_{S_{p_{1:n}} \sim p_{1:n}}\mathbb{E}_{S'_{p_{1:n}} \sim p_{1:n}}\left[\sup_{f \in \mathcal{F}} \frac{1}{n}\sum_{j=1}^{n}\frac{1}{m}\sum_{i=1}^{m}(\ell(f(\vec{x}'_{ij}), y'_{ij}) - \ell(f(\vec{x}_{ij}), y_{ij}))\right] \tag{16}$$

$$= \mathbb{E}_{S_{p_{1:n}} \sim p_{1:n}}\mathbb{E}_{S'_{p_{1:n}} \sim p_{1:n}}\mathbb{E}_{\vec{\sigma}}\left[\sup_{f \in \mathcal{F}} \frac{1}{n}\sum_{j=1}^{n}\frac{1}{m}\sum_{i=1}^{m}\sigma_{ij}(\ell(f(\vec{x}'_{ij}), y'_{ij}) - \ell(f(\vec{x}_{ij}), y_{ij}))\right] \tag{17}$$

$$\leq \mathbb{E}_{S'_{p_{1:n}} \sim p_{1:n}}\mathbb{E}_{\vec{\sigma}}\left[\sup_{f \in \mathcal{F}} \frac{1}{n}\sum_{j=1}^{n}\frac{1}{m}\sum_{i=1}^{m}\sigma_{ij}\ell(f(\vec{x}'_{ij}), y'_{ij})\right] \tag{18}$$

$$+ \mathbb{E}_{S_{p_{1:n}} \sim p_{1:n}}\mathbb{E}_{\vec{\sigma}}\left[\sup_{f \in \mathcal{F}} \frac{1}{n}\sum_{j=1}^{n}\frac{1}{m}\sum_{i=1}^{m}-\sigma_{ij}\ell(f(\vec{x}_{ij}), y_{ij})\right] \tag{19}$$

$$= 2\mathbb{E}_{S_{p_{1:n}} \sim p_{1:n}}\mathbb{E}_{\vec{\sigma}}\left[\sup_{f \in \mathcal{F}} \frac{1}{n}\sum_{j=1}^{n}\frac{1}{m}\sum_{i=1}^{m}\sigma_{ij}\ell(f(\vec{x}_{ij}), y_{ij})\right] \tag{20}$$

$$= 2\mathbb{E}_{S_{p_{1:n}} \sim p_{1:n}}[\hat{\mathcal{R}}_{mn}(\mathcal{F})], \tag{21}$$

where the first inequality is from moving the supremum inside the expectation and the second is from subadditivity of suprema. Observing that the absolute difference of computing $\hat{\mathcal{R}}_{mn}(\mathcal{F})$ on $\mathcal{S}$ and $\mathcal{S}'$

cannot exceed $\frac{1}{mn}$, another application of McDiarmid's inequality tells us that with confidence at least $1 - \frac{\delta}{2}$

$$2\mathbb{E}_{S_{p_{1:n}} \sim p_{1:n}}[\hat{\mathcal{R}}_{mn}(\mathcal{F})] \leq 2\hat{\mathcal{R}}_{mn}(\mathcal{F}) + 2\sqrt{\frac{\ln(2/\delta)}{2mn}}. \tag{22}$$

Combining Equations 12 and 22 with the union bound concludes the proof. $\qquad\square$

### A.1 Proof of Unobservable Bound

*Proof.* We start by bounding the left-hand side of the main theorem statement as

$$L_q(f) = \frac{1}{n}\sum_{j=1}^{n} L_{p_j}(f) + L_q(f) - \frac{1}{n}\sum_{j=1}^{n} L_{p_j}(f) \tag{23}$$

$$\leq \frac{1}{n}\sum_{j=1}^{n} L_{p_j}(f) + \frac{1}{n}\sum_{j=1}^{n} \int_{\mathcal{Z}} f(\vec{z})(q(\vec{z}) - p_j(\vec{z})) \cdot d\vec{z} \tag{24}$$

$$\leq \frac{1}{n}\sum_{j=1}^{n} L_{p_j}(f) + \frac{1}{n}\sum_{j=1}^{n} \|f\|_\infty \|q - p_j\|_1 \tag{25}$$

$$= \frac{1}{n}\sum_{j=1}^{n} L_{p_j}(f) + \frac{2}{n}\sum_{j=1}^{n} \|q - p_j\|_{TV}. \tag{26}$$

$$\tag{27}$$

The result follows by applying Theorem 4 to the first summation. $\qquad\square$

### A.2 Proof of Observable Bound

We now prove Theorem 2.

*Proof.* Theorem 4 tell us that with confidence at least $1 - \delta$,

$$\frac{1}{n}\sum_{j=1}^{n} L_{p_j}(f) \leq \hat{L}^{\mathcal{E}}(f) + 2\hat{\mathcal{R}}_{mn}(\mathcal{F}) + 3\sqrt{\frac{\ln(2/\delta)}{2mn}}. \tag{28}$$

Thus, we must provide a (high confidence) upper bound on

$$L^{\mathcal{E}}(f) - \frac{1}{n}\sum_{j=1}^{n} L_{p_j}(f) \tag{29}$$

that holds uniformly for all $f \in \mathcal{F}$. We can use the same idea the proof for Theorem 4 to show how Rademacher complexity controls generalisation to novel domains, rather than novel instances within the same domain. Begin by letting $P = \{p_1, ..., p_n\}$ be an i.i.d. sample of $n$ domains from $\mathcal{E}$, and define

$$\Phi(P) = \sup_{f \in \mathcal{F}} L^{\mathcal{E}}(f) - \frac{1}{n}\sum_{j=1}^{n} L_{p_j}(f). \tag{30}$$

If we construct $P'$ by replacing any $p_j \in P$ with $p'_j \sim \mathcal{E}$, then we have $|\Phi(P) - \Phi(P')| \leq \frac{1}{n}$, so McDiarmid's inequality tells us that with confidence at least $1 - \frac{\delta}{2}$

$$\Phi(P) \leq \mathbb{E}_{p_{1:n} \sim \mathcal{E}}[\Phi(P)] + \sqrt{\frac{\ln(2/\delta)}{2n}}. \tag{31}$$

We proceed by bounding the expected value of $\Phi(P)$,

$$\mathbb{E}_{p_{1:n}\sim\mathcal{E}}\left[\sup_{f\in\mathcal{F}}\left(\mathbb{E}_{q\sim\mathcal{E}}[L_q(f)] - \frac{1}{n}\sum_{j=1}^{n}L_{p_j}(f)\right)\right] \tag{32}$$

$$= \mathbb{E}_{p_{1:n}\sim\mathcal{E}}\left[\sup_{f\in\mathcal{F}}\mathbb{E}_{q_{1:n}\sim\mathcal{E}}\left[\frac{1}{n}\sum_{j=1}^{n}\left(L_{q_j}(f) - L_{p_j}(f)\right)\right]\right] \tag{33}$$

$$\leq \mathbb{E}_{p_{1:n},q_{1:n}\sim\mathcal{E}}\left[\sup_{f\in\mathcal{F}}\frac{1}{n}\sum_{j=1}^{n}\left(L_{q_j}(f) - L_{p_j}(f)\right)\right] \tag{34}$$

$$= \mathbb{E}_{p_{1:n},q_{1:n}\sim\mathcal{E}}\mathbb{E}_{\vec{\sigma}}\left[\sup_{f\in\mathcal{F}}\frac{1}{n}\sum_{j=1}^{n}\sigma_j\left(L_{q_j}(f) - L_{p_j}(f)\right)\right] \tag{35}$$

$$\leq 2\mathbb{E}_{p_{1:n}\sim\mathcal{E}}\mathbb{E}_{\vec{\sigma}}\left[\sup_{f\in\mathcal{F}}\frac{1}{n}\sum_{j=1}^{n}\sigma_j L_{p_j}(f)\right] \tag{36}$$

$$= 2\mathbb{E}_{p_{1:n}\sim\mathcal{E}}\mathbb{E}_{\vec{\sigma}}\left[\sup_{f\in\mathcal{F}}\frac{1}{n}\sum_{j=1}^{n}\sigma_j\mathbb{E}_{(\vec{x},y)\sim p_j}[\ell(f(\vec{x}),y)]\right] \tag{37}$$

$$\leq 2\mathbb{E}_{p_{1:n}\sim\mathcal{E}}\mathbb{E}_{(\vec{x}_j,y_j)\sim p_j}\mathbb{E}_{\vec{\sigma}}\left[\sup_{h\in\mathcal{F}}\frac{1}{n}\sum_{j=1}^{n}\sigma_j\ell(f(\vec{x}_j),y_j)\right] \tag{38}$$

$$= 2\mathbb{E}_{p_{1:n}\sim\mathcal{E}}\mathbb{E}_{(\vec{x}_j,y_j)\sim p_j}[\hat{\mathcal{R}}_n(\mathcal{F})], \tag{39}$$

where the first inequality comes from moving the supremum inside the expectation, $\vec{\sigma}$ is a vector of Rademacher random variables, the second inequality is due to the subadditivity of suprema, and the final inequality comes from moving the expectation outside of the supremum. Noting that replacing one of the $(\vec{x}_j, y_j)$ pairs in the final equality will result in $\hat{\mathcal{R}}_n(\mathcal{F})$ changing by at most $\frac{1}{n}$, McDiarmid's inequality can be used to say with confidence $1 - \frac{\delta}{2}$ that

$$2\mathbb{E}_{p_{1:n}\sim\mathcal{E}}\mathbb{E}_{(\vec{x}_j,y_j)\sim p_j}[\hat{\mathcal{R}}_n(\mathcal{F})] \leq 2\hat{\mathcal{R}}_n(\mathcal{F}) + 2\sqrt{\frac{\ln(2/\delta)}{2n}}. \tag{40}$$

Combining Equations 28, 31, and 40 using the union bound completes the proof. $\qquad\square$

## B  PROOF OF COROLLARY 1

*Proof.* We begin with the expected excess risk, which can be bounded from above by

$$\mathbb{E}_S[L^{\mathcal{E}}(\hat{f}) - L^{\mathcal{E}}(f^*)] = \mathbb{E}_S[L^{\mathcal{E}}(\hat{f}) - \hat{L}^{\mathcal{E}}(\hat{f})] + \mathbb{E}_S[\hat{L}^{\mathcal{E}}(\hat{f}) - \hat{L}^{\mathcal{E}}(f^*)] + \mathbb{E}_S[\hat{L}^{\mathcal{E}}(f^*) - L^{\mathcal{E}}(f^*)]$$
$$\tag{41}$$

$$\leq \mathbb{E}_S[L^{\mathcal{E}}(\hat{f}) - \hat{L}^{\mathcal{E}}(\hat{f})] + \mathbb{E}_S[\hat{L}^{\mathcal{E}}(f^*) - L^{\mathcal{E}}(f^*)] \tag{42}$$

$$= \mathbb{E}_S[L^{\mathcal{E}}(\hat{f}) - \hat{L}^{\mathcal{E}}(\hat{f})] \tag{43}$$

$$= \mathbb{E}_S\left[L^{\mathcal{E}}(\hat{f}) - \frac{1}{n}\sum_{j=1}^{n}L_{p_j}(\hat{f})\right] + \mathbb{E}_S\left[\frac{1}{n}\sum_{j=1}^{n}L_{p_j}(\hat{f}) - \hat{L}^{\mathcal{E}}(\hat{f})\right], \tag{44}$$

where the inequality arises because, by definition, the *empirical* risk of $\hat{f}$ must be less than or equal to the empirical risk of the optimal model. The second equality comes from the fact that $f^*$ is determined independently of the particular training set that we sample, so $\mathbb{E}_S[\hat{L}^{\mathcal{E}}(f^*)] = L^{\mathcal{E}}(f^*)$. For the final equality: the first term can be bounded from above (with confidence $1 - \delta$) using the bound for the expected value of $\Phi(S)$ derived in the proof of Theorem 4; and the second term can be bounded from above (with confidence $1 - \delta$) using the bound for the expected value of $\Phi(P)$ derived in the proof for Theorem 2. Combining these two high confidence bounds using the union bound yields the result. $\qquad\square$

## C  PROOF OF THEOREM 3

*Proof.* The result follows from Lemma 1 and the following bound on the population variance,

$$\text{Var}_q[L_q(f)] = \text{Var}_{q_{1:n}}\left[\frac{1}{n}\sum_{i=1}^{n}L_{q_i}(f)\right] \tag{45}$$

$$= \mathbb{E}_{q_{1:n}\sim\mathcal{E}}\left[\frac{1}{n}\sum_{i=1}^{n}L_{q_i}^2(f)\right] - \mathbb{E}_{q_{1:n}\sim\mathcal{E}}\left[\frac{1}{n}\sum_{i=1}^{n}L_{q_i}(f)\right]^2 \tag{46}$$

$$\leq \mathbb{E}_{q_{1:n}\sim\mathcal{E}}\left[\frac{1}{n}\sum_{i=1}^{n}L_{q_i}^2(f)\right] \tag{47}$$

$$\leq L^{\mathcal{E}}(f), \tag{48}$$

where the first inequality is from the positivity of the second term, and the second is because $\ell$ takes values in the range $[0,1]$. □

## D  FURTHER ANALYSIS

### D.1  COMPLETE RESULTS FOR DEPENDENCE ON MODEL COMPLEXITY

In Section 4.1 we showed three of six datasets due to space constraint. The full set of resuts across six datasets are shown here in Figure 4.

### D.2  WORST-CASE ANALYSIS

We also consider the worst case scenario in Fig. 5, where the worst out of all held out domains is reported, in contrast to the average case in Fig. 1. From the results, we can see that: (i) worst case performance is also governed by model complexity, and (ii) the best complexity value for worst case metric is very similar to the corresponding complexity for average case, as predicted in Section 3.5.

### D.3  ANALYZING DOMAINBED METHODS

In Section 4.2, we analyzed the performance and complexity of ERM in DomainBed as a function of the a representative hyper-parameter – the number of training iterations. Here we analyze the performance and complexity of several published neural DG methods implemented in DomainBed, run repeatedly with random hyper-parameters. From the plot in Figure 6 we can see that: (i) different hyper-parameters can lead the same model to have different complexity and different accuracy. (ii) While the decay with respect to overfitting is less pronounced here than for linear models in Fig. 1, there is a slow decay past the peak complexity, which is around 1 in each dataset. The reason for the less pronounced decay with higher complexity may be that our measure is an *upper bound* on complexity, with effective complexity likely being lower. Ongoing developments in tighter Rademacher complexity bounds may reveal clearer structure in future. Overall these results provide a possible explanation for the hyper-parameter sensitivity highlighted in (20) – hyper-parameter choice influences complexity and hence generalisation via Thm 2; and a possible explanation for differing performance of state of the art methods – if each method provides a different kind of explicit or implicit regularisation, then they could each find a different empirical risk and complexity tradeoff.

**How Do DG Models Control Complexity?**     By different modifications to standard ERM, DG models explicitly or implicitly modify the bias and variance of the function class to be learned. (20) highlight neural models as being dependent on learning rate, batch size, dropout rate, and weight decay; with other factors being choice of optimiser and weight initialiser, etc. RSC introduce more sophisticated dropout-like mechanisms, which would be expected to reduce complexity. Meanwhile alignment-based methods like CORAL and MMD effectively add auxiliary losses, which will implicitly affect complexity, but are hard to explicitly link to it. Consistency based methods like IRM and VRex penalise loss variance on the training data, which also tends to reduce the generalisation gap in the single task case (27), and may have links to the worst-case analysis in Section 3.4 that depends

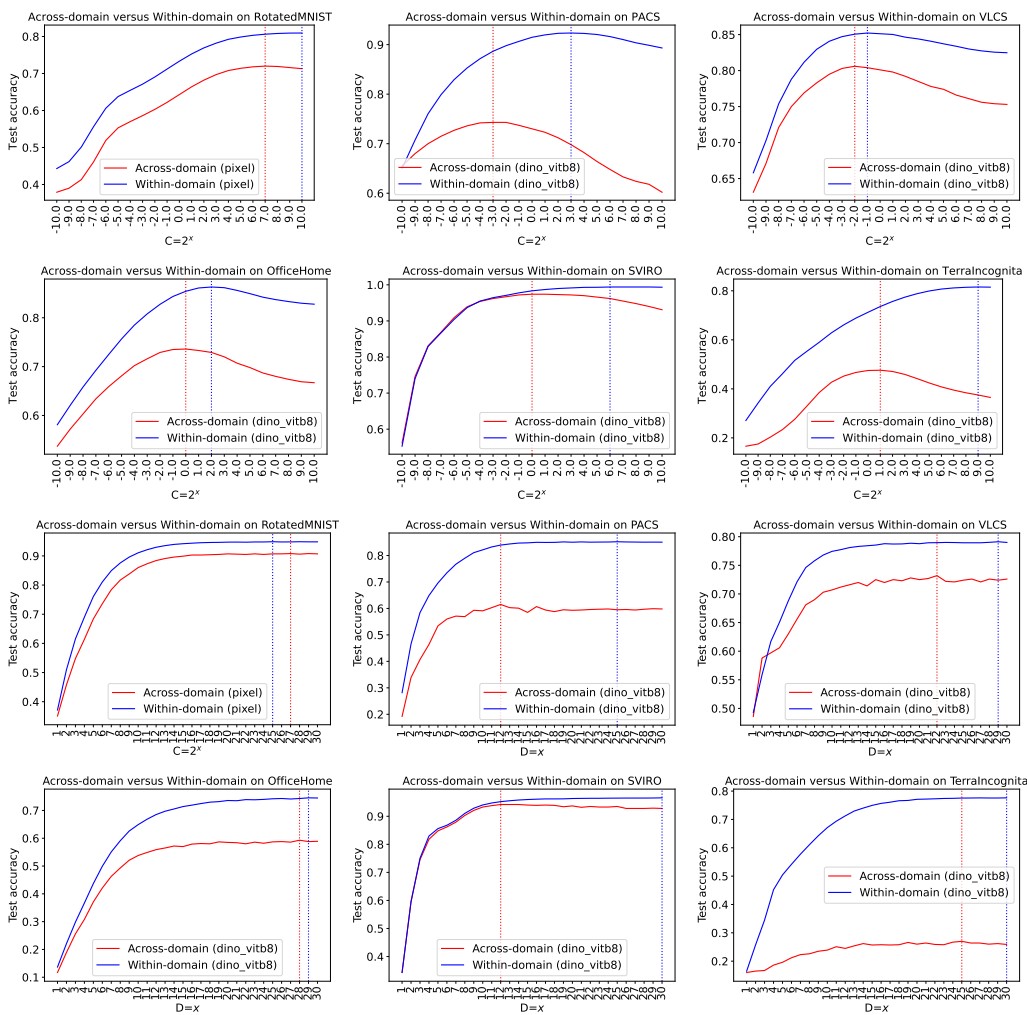

Figure 4: Linear SVM (Top two rows) and Random Forest (Bottom two rows) performance on DomainBed benchmark datasets. Optimal tuning for performance on novel target domains (red) always requires stronger regularisation (lower $C$ or $D$) than for performance on seen domains (blue).

upon the population variance. The specific setting of all the corresponding hyperparameters (e.g., regulariser strength, dropout rates) influence final model complexity, which influences performance.

## E    DETAILED RESULTS

**Detailed results on each DG benchmark**   In Tab. 1 we reported results of our linear model competitors on DomainBed summarised over choice of held out domain. In Tabs. 2-7, we report the detailed results of accuracy and choice of $C$ paramater for each held out domain.

| Cross Validation | | 0 | 15 | 30 | 45 | 60 | 75 | Ave. |
|---|---|---|---|---|---|---|---|---|
| Raw pixel | | | | | | | | |
| Domain Wise | Acc | 0.577 | 0.785 | 0.799 | 0.783 | 0.769 | 0.586 | 0.716 (0.096) |
| | C | 256.0 | 128.0 | 128.0 | 128.0 | 64.0 | 256.0 | |
| Instance Wise | Acc | 0.582 | 0.774 | 0.788 | 0.783 | 0.765 | 0.579 | 0.712 (0.093) |
| | C | 1024.0 | 512.0 | 1024.0 | 1024.0 | 1024.0 | 1024.0 | |

Table 2: Accuracy and selected 'C' on Rotated-MNIST, using LinearSVC.

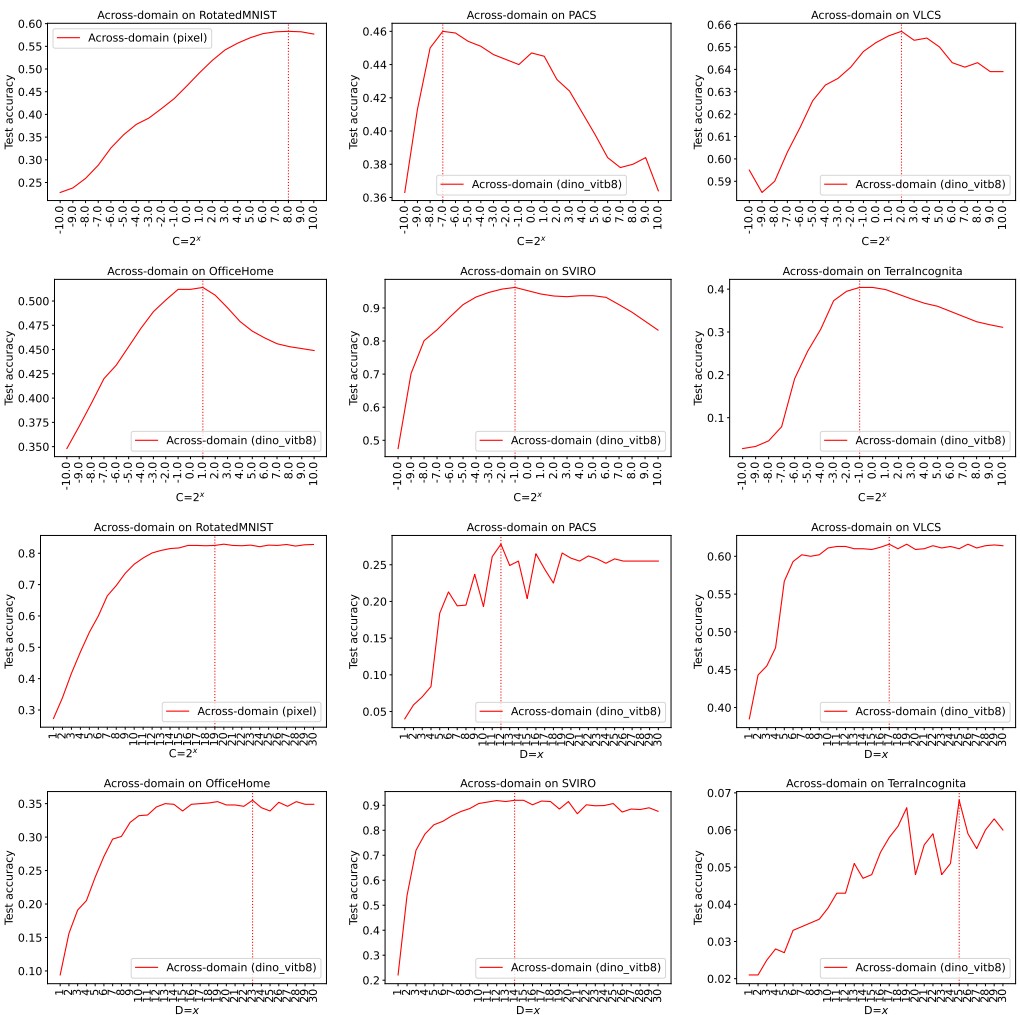

Figure 5: Worst case linear SVM (Top: Row 1-2) and Random Forest (Bottom: Row 3-4) performance on DomainBed benchmark.

| Cross Validation | | A | C | P | S | Ave. |
|---|---|---|---|---|---|---|
| DINO ViT-B8 | | | | | | |
| Domain Wise | Acc | 0.877 | 0.676 | 0.971 | 0.446 | 0.743 (0.201) |
| | C | 0.25 | 0.125 | 0.25 | 0.125 | |
| Instance Wise | Acc | 0.866 | 0.620 | 0.891 | 0.445 | 0.706 (0.184) |
| | C | 4.0 | 8.0 | 8.0 | 2.0 | |
| ResNet-50 | | | | | | |
| Domain Wise | Acc | 0.74 | 0.532 | 0.975 | 0.477 | 0.681 (0.196) |
| | C | 8.00 | 8.00 | 32.00 | 8.00 | |
| Instance Wise | Acc | 0.698 | 0.550 | 0.969 | 0.458 | 0.669 (0.193) |
| | C | 256.00 | 128.00 | 256.00 | 64.00 | |

Table 3: Accuracy and selected 'C' on PACS, using LinearSVC.

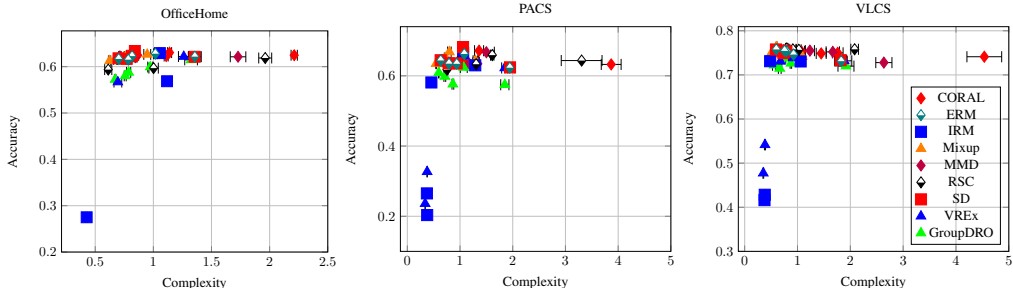

Figure 6: Leave-one-domain out cross-validation performance of neural networks as a function of their measured model complexity after training on DomainBed (DB). The overall results are consistent with a standard bias-variance trade-off (cf. Fig 1 for linear models), with peak performance in the middle of the complexity range. Various neural DG algorithms evaluated by DomainBed (Each method is trained with five different random hyperparameters). Horizontal error bars correspond to the standard deviation of the model complexity measured for each cross-validation iteration.

| Cross Validation | | C | L | S | V | Ave. |
|---|---|---|---|---|---|---|
| DINO ViT-B8 | | | | | | |
| Domain Wise | Acc | 0.972 | 0.648 | 0.787 | 0.811 | 0.804 (0.115) |
| | C | 0.5 | 0.5 | 0.125 | 0.5 | |
| Instance Wise | Acc | 0.977 | 0.648 | 0.785 | 0.811 | 0.805 (0.117) |
| | C | 0.25 | 0.5 | 0.5 | 0.5 | |
| ResNet-50 | | | | | | |
| Domain Wise | Acc | 0.986 | 0.598 | 0.730 | 0.755 | 0.767 (0.140) |
| | C | 16.00 | 8.00 | 8.00 | 1.00 | |
| Instance Wise | Acc | .986 | 0.598 | 0.730 | 0.792 | 0.776 (0.140) |
| | C | 16.00 | 8.00 | 8.00 | 8.00 | |

Table 4: Accuracy and selected 'C' on VLCS, using LinearSVC.

| Cross Validation | | A | C | P | R | Ave. |
|---|---|---|---|---|---|---|
| DINO ViT-B8 | | | | | | |
| Domain Wise | Acc | 0.748 | 0.512 | 0.832 | 0.850 | 0.736 (0.135) |
| | C | 1.0 | 1.0 | 1.0 | 0.5 | |
| Instance Wise | Acc | 0.730 | 0.514 | 0.833 | 0.846 | 0.731 (0.133) |
| | C | 4.0 | 2.0 | 2.0 | 4.0 | |
| ResNet-50 | | | | | | |
| Domain Wise | Acc | 0.681 | 0.521 | 0.784 | 0.813 | 0.700 (0.114) |
| | C | 8.00 | 8.00 | 4.00 | 8.00 | |
| Instance Wise | Acc | 0.652 | 0.513 | 0.794 | 0.803 | 0.691 (0.119) |
| | C | 64.00 | 32.00 | 16.00 | 64.0 | |

Table 5: Accuracy and selected 'C' on OfficeHome, using LinearSVC.

| Cross Validation | | aclass | escape | hilux | i3 | lexus | tesla | tiguan | tucson | x5 | zoe | Ave. |
|---|---|---|---|---|---|---|---|---|---|---|---|---|
| DINO ViT-B8 | | | | | | | | | | | | |
| Domain Wise | Acc | 0.979 | 0.980 | 0.967 | 0.997 | 0.959 | 0.981 | 0.942 | 0.986 | 0.975 | 0.954 | 0.972 (0.016) |
| | C | 1.0 | 1.0 | 1.0 | 1.0 | 2.0 | 1.0 | 2.0 | 1.0 | 1.0 | 2.0 | |
| Instance Wise | Acc | 0.922 | 0.991 | 0.910 | 0.998 | 0.902 | 0.960 | 0.937 | 0.983 | 0.976 | 0.955 | 0.953 (0.032) |
| | C | 128.0 | 64.0 | 128.0 | 64.0 | 256.0 | 128.0 | 256.0 | 128.0 | 128.0 | 128.0 | |
| ResNet-50 | | | | | | | | | | | | |
| Domain Wise | Acc | 0.963 | 0.941 | 0.961 | 0.926 | 0.942 | 0.940 | 0.947 | 0.951 | 0.913 | 0.847 | 0.933 (0.032) |
| | C | 16.00 | 64.00 | 128.00 | 128.00 | 32.00 | 16.00 | 256.00 | 128.00 | 64.00 | 64.00 | |
| Instance Wise | Acc | 0.968 | 0.928 | 0.959 | 0.877 | 0.954 | 0.931 | 0.954 | 0.958 | 0.827 | 0.913 | 0.927 (0.042) |
| | C | 1024.00 | 1024.00 | 1024.00 | 1024.00 | 1024.00 | 1024.00 | 1024.00 | 1024.00 | 1024.00 | 1024.00 | |

Table 6: Accuracy and selected 'C' on SVIRO, using LinearSVC.

| Cross Validation | | L100 | L38 | L43 | L46 | Ave. |
|---|---|---|---|---|---|---|
| DINO ViT-B8 | | | | | | |
| Domain Wise | Acc | 0.444 | 0.457 | 0.516 | 0.399 | 0.454 (0.041) |
| | C | 0.25 | 1.0 | 1.0 | 2.0 | |
| Instance Wise | Acc | 0.472 | 0.391 | 0.366 | 0.324 | 0.388 (0.054) |
| | C | 256.0 | 128.0 | 256.0 | 256.0 | |
| ResNet-50 | | | | | | |
| Domain Wise | Acc | 0.305 | 0.145 | 0.338 | 0.344 | 0.283 (0.081) |
| | C | 4.00 | 2.00 | 16.00 | 32.00 | |
| Instance Wise | Acc | 0.211 | 0.336 | 0.271 | 0.257 | 0.269 (0.045) |
| | C | 1024.00 | 1024.00 | 1024.00 | 1024.00 | |

Table 7: Accuracy and selected 'C' on Terra Incognita, using LinearSVC.

