# OpenReview forum: "An Investigation of Domain Generalization with Rademacher Complexity"
_ICLR.cc/2023/Conference — Submitted to ICLR 2023_

### Official Review · Reviewer_EU2D · 2022-10-25

**Confidence:** 5
**Correctness:** 4
**Technical Novelty And Significance:** 1
**Empirical Novelty And Significance:** 2
**Recommendation:** 3

**Clarity, Quality, Novelty And Reproducibility:**

Clarity: Fine
Quality: Low
Novelty: Poor
Reproducibility: Fine

**Strength And Weaknesses:**

Unfortunately, I think the contribution of this work is extremely limited. First of all, the work situates itself by claiming that previous works make either structural assumptions, or "restrict to specific classes, such as kernel machines", and then purports to do something different. This is incorrect.

In domain generalization, there are really only two options for hoping to be able to generalize. We could (a) assume the domains are drawn from a prior (as this work and several previous works do [1, 2]), or we could (b) make a structural assumption. This work presents option (b) as if it is undesirable, when it is one of only two options and is in fact the more popular one---and the one that most people mean when they use the term "domain generalization" in a modern context. The reason option (b) is more popular is because option (a) is somewhat trivial: just use ERM, with sufficient regularization. This was shown by [1, 2] a *long* time ago to be sufficient (and it is unsurprising), and so the fact that ERM does not work in practice implies that assuming a prior is still missing something important and that option (b) is a possible way to address this.

Now, *conditioning on the use of option (b)*, this work doesn't really make any contributions. [1, 2] already study the setting of a prior over domains. They don't make a "kernel machine modeling assumption", they derive bounds on the OOD risk which uses RKHS norms. What does this work do instead? They apply standard Rademacher bounds to the prior over environments. Obviously, the risk of a function class $\mathcal{F}$ converges uniformly to its expected risk with respect to the distribution of environments, so by assuming this prior we can just plug in existing Rademacher bounds, which is exactly what this work does. The "assuming a prior" idea (i.e., option (b)) is not new, and there is no mathematical contribution. Replacing the RKHS norm with the Rademacher complexity of neural networks is not an improvement, it's just another option.

I don't think the experiments really show much either, but even if they did, 3/4 of the paper is devoted to what amounts to plugging existing bounds into a known setting. Unfortunately I don't think this is a matter of making changes to the draft---the premise of the paper as it currently stands just doesn't really say anything new.

[1] Generalizing from Several Related Classification Tasks to a New Unlabeled Sample. Blanchard et al. 2011

[2] Domain Generalization by Marginal Transfer Learning. Blanchard et al. 2017

**Summary Of The Paper:**

The paper studies domain generalization where the domains are drawn from a prior and applies standard rademacher bounds.

**Summary Of The Review:**

See above

---

### Official Review · Reviewer_ss4Y · 2022-10-29

**Confidence:** 3
**Correctness:** 4
**Technical Novelty And Significance:** 2
**Empirical Novelty And Significance:** 2
**Recommendation:** 3

**Clarity, Quality, Novelty And Reproducibility:**

Clarity: I thank the authors for writing a clear paper. I found it easy to read almost the entire paper.
Quality: The experiments are generally well explained, conclusions substantiated, and theoretical bounds reasonably clear (I did not review the proofs).
Originality: I believe this work is just a straight forward application of standard uniform convergence and generalization guarantees of supervised learning. Therefore, I would rate the paper low on originality.

**Strength And Weaknesses:**

Strengths:
(a) The paper studies an important problem of domain generalization and tries to characterize the effect of hyperparameters on the final performance on an unseen domain
(b) The paper is generally well written and easy to follow
(c) The experimental conclusions are well-substantiated

Weaknesses:
My main concern with the paper is regarding its foreseeable impact: the theoretical contributions seem like straightforward generalization of the standard uniform convergence analysis for generalization in distribution and there are new tools or insights. It would help if the authors clarify the main theoretical contributions and novelty/significance.

From a practical perspective, it seems almost obvious that lower capacity would generalize better and that one can try to do cross validation using a held-out domain. The main challenges in practice are: we do not know a good complexity measure of deep networks and this is a big open problem in standard supervised learning. Could the authors please clarify how their notions of complexity are different from those in standard supervised learning?

The experiments provided on linear networks, shallow networks and RBFs make  but how do they compare to the state-of-the-art results with deeper models? If simpler models with appropriate regularization beat claimed state-of-the-art methods with larger models, then this result is interesting. Otherwise, I wonder what takeaways could we use for further development (it seems obvious to me that "regularization" will help, but what's the right regularization is the question which this paper fails to answer).

I agree with the discussion on page 5 that you probably need more assumptions on the different domains in order to get a stronger result, but I think that's the actual interesting regime. Otherwise domain generalization just reduces to standard generalization problem.


**Summary Of The Paper:**

This paper studies the problem of domain generalization where we are given data from multiple domains, and a model has to work well on an unseen domain. The paper theoretically bounds the performance on the unseen domain via a uniform convergence bound involving the Rademacher complexity of the classifier. I see this result as analagous to standard generalization bound, but with IID samples replaced by IID environments. This bound tells us that smaller capacity models are more likely to generalize better to unseen domains than higher capacity models. The paper then verifies this experimentally on linear networks over pretrained features, or two-layer networks, that model complexity (in addition to model fit) plays an important role in determining performance on new domains, in a manner that mirrors the bias-variance tradeoff.

**Summary Of The Review:**

The paper studies an important question of domain generalization, but ends up studying an uninteresting regime (with no assumptions on the data generating process) leading to pretty unsurprising conclusions with no clear actionable takeaway. It would be very valuable if the authors had some discussion on how these results (both theoretical and empirical) inform us on what to do, and how they would help advance the state-of-the-art in domain generalization. In particular, how does the notion of complexity in domain generalization differ from standard supervised learning? If we haven't solved the latter, what hopes are for the former?

---

### Official Review · Reviewer_ECRP · 2022-10-30

**Confidence:** 3
**Clarity, Quality, Novelty And Reproducibility:** 1. For the proof of theorem 2 eq(30),…
**Correctness:** 3
**Technical Novelty And Significance:** 1
**Empirical Novelty And Significance:** 2
**Recommendation:** 3

**Strength And Weaknesses:**

This paper has clearly stated its objective, the approach and the corresponding evaluations. Most part of the paper is easy to follow.

1. I’m worried the contribution is not significant. The technical analysis in terms of Rademacher complexity is rather standard.
2. Some discussion needs to be clarified. For example, in the discussion of corollary 1, I do not completely follow why ``any future DG method that claims to be better than ERM should demonstrate a faster convergence rate than this — at least by an improved constant factor’’. Since you are not providing a lower bound on the excess expected risk, and your upper bound is also in Big-O notation,  what does it mean by an improved constant factor?
3. I believe the worst-case scenario is more interesting for domain generalization even requires mild assumption. Yet the bound provided here (Lemma 1 and Theorem 3) gives us vacuous bounds that do not have any insights.
4. I’m suspicious about the tradeoff proposed by the author. The author makes that assumption based on theorem 2, but it’s unclear whether the bound is tight or not, and the Rademacher complexity bound may be vacuous in reality. The author makes an analogy with the standard overfitting-underfitting trade-off, yet recent experiments observe the double descent phenomena. Thus I’m wondering whether the same double descent phenomena happens in the domain generalization scenario, and how overparameterized model performs in the domain generalization scenario.

**Summary Of The Paper:**

The author provides domain generalization bound from Rademacher Complexity aspect and shows there’s a tradeoff between empirical risk and model complexity in domain generalization. The author provides experimental results to support their theoretical findings.

**Summary Of The Review:**

Overall I’m not convinced by the findings of the paper. The technical proof idea is not novel, and the contribution is not significant. The writing of some parts of the paper I believe needs to be improved.

---

### Official Review · Reviewer_rAHY · 2022-10-31

**Confidence:** 3
**Correctness:** 4
**Technical Novelty And Significance:** 2
**Empirical Novelty And Significance:** 2
**Recommendation:** 3

**Clarity, Quality, Novelty And Reproducibility:**

The paper is written clearly with reasonably good quality. As far as I know, the work is original.

**Strength And Weaknesses:**

Strength
- As far as I know, the problem of considering a distribution of i.i.d. domains is novel.
- The paper is clearly written and every theorem is followed by detailed discussions.
- The experiments, despite only using simple models, are very extensive. Many datasets are used, and experiment details are clearly stated.

Weakness
- My main criticism is that, to some extent, the paper is cheating in the sense that the testing data is not really o.o.d, because the domains are assumed to be sampled i.i.d. from the “distribution of domains”. My feeling is that the authors are simply rephrasing an in-domain generalization problem and making it look like an out-of-domain generalization problem. In fact, suppose m=1, then the problem recovers exactly the same problem of in-domain generalization.
- Also, from the bound it looks like n (number of observed domains) does need to be very large to make the bound non-vacuous, which is not practical at all. In most domain generalization problems, we only have one source domain.
- There’s very limited technical novelty. The proof is simply reusing the standard Rademacher complexity bound but with simple adjustment to the two levels of i.i.d. samples (sampling of the domain and sampling of data within a domain).
- The experiments are all using simple models like linear models or random forest. It’s unclear how these results are relevant in the deep learning era.

**Summary Of The Paper:**

This paper gives a few generalization bounds in the domain generalization setting. They assume there are n domains that are i.i.d. sampled from some “distribution of domains”, and for each domain there are m i.i.d. sampled datapoints. In this case, they provide a generalization bound where the Rademacher complexity depends on both n and m. Furthermore, they provide a high probability bound for the worst domain in the “distribution of domains” using standard Chebyshev’s inequality. They justify the practical relevance of their method with some experiments (with simple models like svm and random forest).

**Summary Of The Review:**

It think the paper is not really considering OOD problems as the authors claim. Also, the technical novelty is very limited as far as I can tell. Thus, I would say this is below the bar of acceptance.

---

### Decision · Program_Chairs · 2023-01-20

**Decision:**

Reject

**Justification For Why Not Higher Score:**

The weaknesses are critical and unanimous. Moreover, authors did not submit a rebuttal.

**Justification For Why Not Lower Score:**

N/A

**Metareview: Summary, Strengths And Weaknesses:**

The paper is extending existing Rademacher based generalization bounds to the domain generalization case. The main assumptions is domains being sampled iid. from some stationary domain distribution. All reviewers agree that i) the results are largely straightforward extensions of existing bounds and ii) they are not adding anything interesting on top of existing results (like Blanchard et al. 2011, 2017). Moreover, authors did not submit a rebuttal.